# Areas of Uncertainty in SARS-CoV-2 Vaccination for Cancer Patients

**DOI:** 10.3390/vaccines10122117

**Published:** 2022-12-11

**Authors:** Anastasios Dimou

**Affiliations:** Division of Medical Oncology, Mayo Clinic, 200 1st St SW, Rochester, MN 55905, USA; dimou.anastasios@mayo.edu

**Keywords:** T cell assay, cellular immunity, humoral immunity, neutralizing antibodies

## Abstract

Early in the COVID-19 pandemic, it was recognized that infection with SARS-CoV-2 is associated with increased morbidity and mortality in patients with cancer; therefore, preventive vaccination in cancer survivors is expected to be particularly impactful. Heterogeneity in how a neoplastic disease diagnosis and treatment interferes with humoral and cellular immunity, however, poses a number of challenges in vaccination strategies. Herein, the available literature on the effectiveness of COVID-19 vaccines among patients with cancer is critically appraised under the lens of anti-neoplastic treatment optimization. The objective of this review is to highlight areas of uncertainty, where more research could inform future SARS-CoV-2 immunization programs and maximize benefits in the high-risk cancer survivor population, and also minimize cancer treatment deviations from standard practices.

## 1. Introduction

The ongoing global pandemic of severe acute respiratory syndrome coronavirus 2 (SARS-CoV-2) has caused over half a billion infections and six million deaths as of the time of this writing (https://coronavirus.jhu.edu (accessed on 15 August 2022)). Early in the pandemic, a diagnosis of cancer was recognized as one of the top risk factors for morbidity and mortality from the infection [1,2,3]. Additionally, the ramifications of the global response to the pandemic encompass the entire spectrum of cancer care, including delays in diagnosis, screening deferral, treatment interruptions, psychological stress from social distancing, and hurdles to access to innovative clinical trials.

The β coronavirus family includes the Middle East respiratory syndrome (MERS) coronavirus, SARS-CoV, and SARS-CoV-2. SARS-CoV-2 is a single-stranded RNA virus with an envelope that contains glycosylated spike proteins (S) that the virus uses to bind angiotensin-converting enzyme-2 (ACE-2) and enter host cells. Cleavage of the spike protein by transmembrane serine protease-2 (TMPRSS2) generates the fragments S1 and S2. The former fragment contains the receptor binding domain that binds ACE2 and facilitates viral cellular entry via fusion between the viral and cellular membranes with the release of viral RNA in the host cell cytoplasm [4]. Alternatively, the virus can enter the host cells via the endocytic pathway that does not involve TMPRSS2 [4]. Other structural proteins besides S include the M protein, which is the most abundant transmembrane protein; the N protein, which encapsulates the viral RNA; the E protein, which aids in viral assembly. Additionally, SARS-CoV-2 contains a series of non-structural and accessory proteins [5]. Many of these proteins inhibit type I and type III host IFN responses [6]. Typical symptoms include fever, sore throat, malaise, diarrhea, anosmia, and ageusia [7,8,9], and manifestations range from asymptomatic to end organ damage, including but not limited to the lungs [10]. Arterial and venous thrombosis can occur in more severe cases [11]. In addition to active cancer and recent use of antineoplastics, other well-recognized risk factors for severe disease include male sex, older age, obesity, immunosuppression, cardiovascular disease, and comorbidities [12]. Symptoms such as fatigue and headaches can persist in a group of patients after the acute infection [13].

A number of vaccines have been approved by regulatory agencies for use in the general population after placebo-controlled studies showed that vaccination prevents infection and complications or death from SARS-CoV-2. Among them, the mRNA 1273 and the BNT162b2 use lipid nanoparticle-encapsulated nucleoside-modified RNA that encodes the spike protein [14,15]. Other vaccines include the Ad26.COV2.S, a recombinant, replication-incompetent human adenovirus type 26 vector which encodes the spike protein; the ChAdOx1 nCoV-19, a replication-deficient chimpanzee adenoviral vector containing the sequence for the SARS-CoV-2 structural surface glycoprotein antigen; the NVX-CoV2373, a recombinant nanoparticle spike protein and adjuvant [16,17,18]. Additionally, a series of inactivated whole virus vaccines have been developed [19]. VLA2001 is an adjuvanted inactivated whole-virus vaccine that was shown to induce higher neutralizing antibodies compared to the ChAdOx1 nCoV-19 vaccine in a phase 3 immunobridging trial [20]. These developments occurred at an unprecedentedly fast pace compared to the time that vaccines have historically become available to the general population. The early success with vaccines against SARS-CoV-2 notwithstanding, cancer patients on active antineoplastic treatment were excluded from the seminal studies that led to the original regulatory approvals. Smaller studies established the safety of those vaccines in the cancer population. An increasing body of literature has established that mRNA-based and other vaccines effectively raise antibody responses in patients with cancer, albeit at lower titers than in individuals without cancer [21,22,23,24].

A hallmark of the SARS-CoV-2 pandemic has been the emergence of variants of concern (VOCs) with mutations in the spike protein that dominated different waves of the pandemic. VOCs have variable ability to utilize the TMPRSS2 dependent versus the endocytic pathway for viral entry in the host cell, which has an impact on the types of target cells that are most vulnerable to viral infection. Specifically, S protein cleavage in the B.1.1.529 (omicron) VOC that emerged in 2021 is not as efficient as in the B.1.617.2 (delta) VOC. Consequently, this property favors omicron viral entry in cells with low expression of TMPRSS2 and leads to a milder clinical syndrome with predominantly upper respiratory symptoms [25]. Importantly, the vaccines were designed based on the genomic sequence from the index virus and their effectiveness is variable against VOCs. In a recent case control study from the UK, the prevention of symptomatic disease was higher for the delta than for the omicron VOC and effectiveness waned over time [26]. At the time of this writing, the effectiveness of vaccination and boosters at preventing severe disease and death from the omicron VOC over time is not known; however, the ChAdOx1, nCoV-19, and BNT162b2 maintained their ability to prevent hospitalization from infection with the delta VOC for at least six months [26]. On the other hand, the ability of inactivated whole virus vaccines to induce immunity, targeting viral proteins other than the S protein, might indicate that those vaccines could be effective against the VOCs.

Given the challenges with SARS-CoV-2 infection in patients with cancer, as well as the heterogeneous effect of cancer treatments on humoral and cellular immunity, it is important to clarify key areas where future research in this population can be the most impactful. In this review, the critical literature on vaccine effectiveness in cancer patients is summarized and three areas of uncertainty are pointed out. First, gaps in knowledge regarding humoral immune responses following vaccination for SARS-CoV-2 or infection in the cancer patient population are highlighted. Second, the effectiveness of cellular immunity to prevent severe disease and complications from SARS-CoV-2 VOCs in cancer patients within adequate, or waning, humoral immunity is reviewed. Finally, the need to delay or modify antineoplastic treatment and the optimal duration of interruption for patients with cancer who have been vaccinated is discussed.

## 2. Areas of Uncertainty

### 2.1. Humoral Immune Responses to SARS-CoV-2 Vaccines in Patients with Cancer

A number of observational studies indicated that patients with cancer, especially those with hematological malignancies or those receiving B cell depleting therapies, have reduced or no antibody responses to mRNA-based vaccination for SARS-CoV-2. A longitudinal study over six months looked at IgG antibodies against the spike protein in patients with solid tumors and hematological malignancies following vaccination with mRNA-1273 or BNT162b2 vaccines [27]. Peak levels of seropositivity were lower for patients with solid tumors and were the lowest for patients with hematological cancers and, particularly, multiple myeloma, compared to healthy control reference. Antibody levels decreased at sustained time points over six months, with healthy controls having the highest and patients with hematological cancers having the lowest titers. Antineoplastic therapy was associated with a lower seropositivity rate in multivariate models. Antibody levels were higher for patients who received the vaccine at least four weeks after anti-CD20 or at least two weeks after anti-CD38 antibody treatments. Intriguingly, antibody titers were lower at peak and sustained time points for patients who were vaccinated following immune check point inhibitors. Antibody titers against the spike protein, as well as seroconversion rates after vaccination with mRNA-based vaccines, were lower for patients with hematological malignancies and solid tumors on treatment compared to immunocompetent controls in a retrospective cohort from Mayo Clinic that mainly included patients with chronic lymphocytic leukemia or breast cancer [28]. Other studies have reached similar conclusions [29].

Despite these important data, a number of unanswered questions remain. In the study by Figueiredo et al. [27], the range of seroconversion and antibody titers was broad, indicating significant heterogeneity in humoral responses. Confounding effects of booster vaccination doses, as well as pre-existing immunity from prior infection with SARS-CoV-2 or other coronaviruses, could explain some of the heterogeneity regardless of cancer history. The timing of vaccination with respect to antineoplastic treatment also requires further study. Treatment with anti-CD20 agents has been repeatedly shown to eliminate the ability to raise neutralizing antibodies following vaccination or infection for up to six months. However, these treatments do not affect pre-existent immunity and, therefore, vaccination should ideally be offered prior to treatment start [30]. The link between immune check point inhibition and lower antibody responses to vaccines is important and would need validation. Frequently, patients on cancer immunotherapy receive treatment with corticosteroids and other immunosuppressive agents to address toxicities. Additionally, there is mechanistic evidence to support immune check point inhibition itself as a direct factor for weakened responses to vaccines for SARS-CoV-2 [31,32]. It is important to clarify the extent to which cancer immunotherapy directly hinders humoral immunity, irrespectively of immunosuppressive treatment for immune related side effects, as well as the effects of cancer immunotherapy on memory B cells and recall antibody responses. In individuals without cancer, recall immunity has been described with reference to VOCs [33,34], and relevant studies in the cancer population are lacking.

### 2.2. Cellular Immune Responses to SARS-CoV-2 Vaccines in Patients with Cancer

Natural infection with SARS-CoV-2 or vaccination induces antibodies that neutralize key proteins in the viral envelope. Importantly, the vaccines are designed to raise humoral responses against the spike protein of SARS-CoV-2. However, mutations in the spike protein during viral replication increase variant fitness to evade neutralizing antibodies. The effectiveness of vaccines at preventing infection following exposure to SARS-CoV-2 delta or omicron VOC is reduced for vaccines that were designed based on the index virus. On the other hand, both natural infection and vaccination also induce adaptive immunity that is maintained by memory T cells. Intriguingly, pre-existing T cell-based immunity was detected in 28–50% of individuals with no prior infection or vaccination history [35].

Sette et al. suggested distinct kinetics of immunological response to SARS-CoV-2 that correlate with the clinical course of the disease [36]. In their model, a delay in innate immunity early in the infection allows the virus to replicate and generate the initial viral load. In most patients, T cell- and antibody-based adaptive immunity will eventually clear the infection. However, a critical delay in adaptive immunity surge is accompanied by uninhibited viral replication and compensatory overwhelming activation of innate immunity in patients with a more severe course. Vaccines bridge the time gap for rapid and effective adaptive immunity in the case of an infection. Among the pillars of adaptive immunity, CD4+ T cells have a predominant role, whereas antibody-producing B cells and CD8+ T cells are also important. Although the development of neutralizing antibodies has been the focus for vaccine effectiveness through mechanisms of opsonization and antibody-dependent cytotoxicity, the increasing body of literature supports cellular immunity as the cornerstone of vaccine activity, especially against variants. It is noteworthy that responses to BNT162b2 and mRNA-1273, or the ChadOx1, nCoV-19 vaccines have a limited neutralizing capacity for the omicron variant; however they still largely reduce hospitalization or death rate from this VOC by more than 70% [37]. Cellular immunity has been suggested to account for this protective effect [38,39]. Guo et al. [40] reported that memory T cells from previous infection retained effectiveness against SARS-CoV-2 variants in vitro for 12 months after initial infection. On the contrary, at the same time point, antibodies poorly neutralized the variant spike protein. Tarke et al. described a multitude of epitopes on SARS-CoV-2 that bind to HLA class I and II alleles to elicit adaptive cellular immune responses that are difficult to overcome by variant mutations [39].

The importance of cellular immunity following vaccination for SARS-CoV-2 to prevent severe disease in patients with cancer and patients who receive antineoplastic treatments is less-well studied. The diagnosis of hematological cancer, as well as treatment with B cell depleting drugs, are well established predictive factors for SARS-CoV-2 morbidity and mortality, paired with a compromised ability of the hosts to neutralize the virus with antibodies [41,42]. It is encouraging that patients with multiple sclerosis who received ocrelizumab, an anti-CD20 depleting therapy, had positive T cell responses despite infrequent positive antibody responses [43]. In a separate report, Apostolidis et al. found a skewed cellular response to mRNA SARS CoV-2 vaccines among patients with multiple sclerosis who were receiving CD20-depleting treatments with enhanced CD8+ T cell induction, preserved type 1 helper T cells (TH1), and compromised circulating follicular helper (T_FH_) cell responses [44]. Given the link between cancer immunotherapy and lower antibody titers, it will be informative to investigate the effects of these treatments on T cell responses.

In cancer patients, the CAPTURE study prospectively studied the humoral and cellular adaptive responses to two doses of the BNT162b2 or AZD1222 vaccines administered 12 weeks apart [45]. Patients with hematological cancers had lower rates of seroconversion and neutralizing antibodies against variants, whereas treatment with anti-CD20 had undetectable antibody responses. Interestingly, T cell responses were intact in 80% of the patients regardless of cancer type (solid tumor or hematological). On the contrary, Ehmsen et al. reported lower rates of cellular responses to mRNA-based vaccines against SARS-CoV-2 among patients with solid (46%) and hematological (45%) cancers [46]. It is noteworthy that any corticosteroid use was associated with lower T cell responses, and more patients in this study were reported to use corticosteroids compared to the CAPTURE study. In CAPTURE, a third of the patients had a history of previous infection with SARS-CoV-2, and these patients had more robust immune responses. Nevertheless, the variability and quality of cellular immune responses in patients with cancer is not well studied, nor have these responses been correlated with clinical outcomes for this patient population. Variability in measuring cellular immunity might further explain the divergent results. Contrary to neutralizing antibody assays, a standardized test to uniformly assess T cell immunity is lacking. The estimation of anti-SARS-CoV-2 immunity at the individual and population level has largely depended on the measurement of antibody titers, mostly with specificity to the spike protein of the virus. However, with the majority of people at the global scale having been exposed to the virus and/or a vaccine, there is great interest in capturing CD4+ and CD8+ T cell-orchestrated immune responses with reproducible and accurate assays that can be applied clinically in real time. These immune responses might be longer-lasting compared to antibody-driven responses and might more reliably predict disease severity from SARS-CoV-2 VOCs. The interferon gamma release assays (IGRAs) are used as blood-based in vitro diagnostics for the *Mycobacterium Tuberculosis* spectrum of disease by measuring IFNγ production following exposure of lymphocytes to *M tuberculosis* antigens [47]. IGRAs have also been proposed for the evaluation of cytomegalovirus-specific CD8+ T cell reconstitution in pediatric patients who undergo hematopoietic stem cell transplantation [48], as well as for the evaluation of cellular immunity response to Varicella Zoster Virus vaccination in individuals with suboptimal IgG responses [49]. In COVID-19 infection, a few reports have shown that the measurement of T cell responses to SARS-CoV-2 with IGRAs is feasible following vaccination [50,51], even in B-cell-depleted patients [52]. A similar methodology could be validated for patients with cancer to predict vulnerability to severe infection from SARS-CoV-2 and guide personalized prevention strategies. The existing cellular immunity tests for SARS-CoV-2 are based on ELISpot or intracellular cytokine staining and are sensitive and accurate; however, they are also time consuming and expensive, and further optimization is an unmet need [53].

### 2.3. Cancer Treatment Modifications in Patients with SARS-CoV-2 Infections

#### 2.3.1. Surgery

Increased morbidity and mortality from SARS-CoV-2 infection in patients with cancer early in the pandemic necessitated the modification of treatment protocols with interruptions and delays in systemic treatments, radiation courses, and surgeries. In a single institution study [54] that looked into the 90-day SARS-CoV-2 infection rates and mortality following surgical treatment for lung cancer, 5 of 41 patients who received lung resection immediately before or in the first few months of the pandemic became infected. Strikingly, of the five infected patients, two died (40%), whereas no deaths were reported in the patients who did not have a SARS-CoV-2 infection diagnosis. In a larger multicenter, international study [55], out of 3778 patients who underwent surgery for treatment of gynecological cancer in early 2020, only 22 were diagnosed with SARS-CoV-2 infection. However, morbidity and mortality were considerably higher in the infected compared to the non-infected population (63.6% vs. 19.1% and 18.2% vs. 0.7%, respectively). Delays in surgical management were documented in 11.2% of the patients, with a detrimental effect on cancer-related prognosis as a result. Increased mortality and morbidity in the post-operative period were observed following surgery for hepatobiliary cancer [56]. Diagnosis of cancer was associated with increased post-operative 90-day mortality in a large study in the UK [57]. In other studies, post-operative diagnosis of SARS-CoV-2 infection was not associated with increased mortality or morbidity, indicating that the type of cancer and surgery, along with other factors, might shape the risk for adverse outcomes [58,59]. It is noteworthy that the application of strict surgical protocols for the avoidance of infection from SARS-CoV-2 was preventive of SARS-CoV-2-related morbidity and mortality from surgery for cancer [57]. On the other hand, treatment practices have changed during the pandemic, with deferral of surgical management and increased use of neoadjuvant therapy in many cases [55,60].

#### 2.3.2. Radiation Therapy

Radiation therapy interruptions are detrimental for cancer prognosis, given the ability of the cancer cells to repopulate during missing fractions. Ying et al. reported that, among patients who interrupted definitive radiation therapy for solid cancer diagnosis, those who discontinued radiotherapy had increased mortality (42.9%) compared to those who completed the radiotherapy despite interruption (6.2%) [61]. An increased frequency of hypofractionated radiation protocols was observed in UK [62]. For patients undergoing head and neck radiotherapy, consensus recommendations by the American Society of Radiation Oncology (ASTRO) and the European Society for Radiotherapy and Oncology (ESTRO) indicate to continue radiation therapy for SARS-CoV-2 infection with mild symptoms and interrupt until recovery from the infection for more severe cases [63]. For patients with lung cancer, on the other hand, while there was strong consensus to delay the start of treatment for SARS-CoV-2 diagnosis, only 57% of the panelists agreed to interrupt radiotherapy for infection diagnosed after the start of treatment and through recovery [64].

#### 2.3.3. Systemic Therapy

In a large study from France that was undertaken early in the pandemic, a SARS-CoV-2 diagnosis led to the interruption or discontinuation of systemic cancer treatment in 39% of the 1092 cases [65]. Interestingly, prior treatment was not associated with increased mortality, except for a small increase in mortality for patients who were admitted to the ICU and had received cytotoxic chemotherapy within the four weeks preceding the infection. In another study from the pre-vaccination phase of the pandemic, systemic therapy safety was similar for solid cancer patients with mild SARS-CoV-2 infection and those where no such infection was reported. Especially, treatment with immune check point inhibitors or other antibody-based drugs with long half-lives prior to SARS-CoV-2 infection was not associated with increased mortality in a large pan cancer study [66]. However, treatment was delayed more frequently for the former group, mainly due to hematological toxicity [67]. At this point, given an increased morbidity and mortality from SARS-CoV-2 infection in patients who receive antineoplastics, and to avoid interactions with antiviral drugs, general practices include the interruption of anticancer systemic therapy in the case of active SARS-CoV-2 infection. The importance of safety notwithstanding, the range of clinical presentation in the case of a positive test for SARS-CoV-2 is broad and includes asymptomatic carriage, mild infection, severe infection with complications, and death. Additionally, a large proportion of patients with cancer receive oral therapies, and it is unclear whether those should be interrupted in the face of asymptomatic or mild infection with the virus. The continuation of treatment with the small molecule tyrosine kinase inhibitor (TKI) alectinib and the TKI crizotinib was possible in two patients with ALK and ROS1 rearranged non-small cell lung cancer, respectively, during infection from SARS-CoV-2 and associated interstitial lung disease [68].

## 3. Concluding Remarks

The SARS-CoV-2 pandemic has had a disproportionately negative impact on the quality of life and prognosis of patients with cancer and their families. The cancer survivor population has heterogeneous vulnerability to the infection that largely depends on the variability of immune system deficits imposed by the underlying malignancy, the various antineoplastics that are administered, and supportive treatments such as corticosteroids. Similarly, protective immunity that is induced from vaccination or natural infection with SARS-CoV-2 can be variable in patients with solid tumors or hematologic malignancies, as has been previously demonstrated for the seasonal flu [69] Therefore, there is an unmet need for the personalization of preventive SARS-CoV-2 vaccination strategies and cancer care modifications in the unfortunate case of a breakthrough COVID-19 infection. Further research on the role of various T cell populations in shaping this risk as well as the standardization of existing tests to capture T cell responses in everyday clinical practice could maximize the benefits from vaccines and minimize deviations from optimal cancer treatment practices.

## Data Availability

Not applicable as there are no original data presented in this review article.

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
