# Peer review of "Areas of Uncertainty in SARS-CoV-2 Vaccination for Cancer Patients"

_vaccines, 2022, doi:10.3390/vaccines10122117_

Round 1
Reviewer 1 Report
The manuscript by Dimou is a short, but helpful summary regarding the effectiveness of COVID-19 vaccines in cancer patients and several issues which are currently unresolved in this respect. The overall reaction of this reviewer is positive. However, the manuscript should be slightly reorganized and some wording, especially in the title and the abstract must be changed in order to avoid potential for confusion or even misrepresentation.
In particular,
1) Words ‘practical implications for future research’ should be dropped from the title since not much of ‘practical implications’ is discussed in the manuscript text.
2) Words ‘patients with solid tumors and hematological malignancies’ in the abstract should be replaced with ‘cancer patients’ since again, mostly no distinction between different malignancies is made in the text (while the discussion of vaccination in patients with multiple sclerosis, a non-malignant disease, is curiously included in lines 156-161).
3) At the end of the introduction, the author promises to discuss ‘3 areas of uncertainty,’ lists them all and then immediately starts Section 2 with the area (Humoral immune responses) that has not been included in this list.
4) Of these, the 2nd ‘area of uncertainty,’ according to the author is the absence of ‘a reliable test to reproducibly measure cellular immunity [against COVID-19] in real time at the bedside.’ However, there is nothing COVID- or cancer-specific to this, such tests are generally unavailable for nearly all current vaccines. I would replace this ‘area’ with the ‘humoral responses’ especially since significant part of the manuscript is dedicated to them. It is also telling that the author had to violate his own subsection structure to include an unnumbered section entitled ‘T cell biomarker to predict disease severity,’ which has no relevant data discussion and can be easily dropped altogether.
5) Section 2.3.4 ‘Cancer treatment modifications in vaccinated patients with breakthrough infections’ should be shortened and moved to ‘Concluding remarks’ since it yet again, has no data analysis whatsoever.
Minor comments.
1) Since there is a single author, he should use ‘I’ instead of royal ‘we’ throughout the manuscript and
2) After diligently reworking the manuscript per comments above, should reread it once again to eliminate strangely sounding sentences like ‘Timing of vaccination with antineoplastic treatment also requires clarification’ (Timing of vaccination together with treatment? Its timing with respect to treatment? Scheduling of vaccination during the treatment?) or ‘The effectiveness of vaccines to prevent infection following exposure to SARS-CoV-2 delta or omicron VOC is reduced for vaccines that were designed based on the wild type virus’ (Is delta or omicron not wild-type? Are we to think that they laboratory-created?). There is a lot of sentences like these, especially towards the end of manuscript.
3) Also, strict adherence to scientific terminology is required, e.g., the author should not talk about ‘diversity of cellular responses’ (line 176), but its ‘variability’ or ‘heterogeneity’ (the same ‘diversity’ term is used on lines 272-273). Similarly, it’s not that ‘those treatments do spare pre-existent immunity’ (lines 106-107) but ‘those treatments do not affect pre-existing’, etc. Attention to these details will be greatly appreciated by readers.
Author Response
I would like to thank reviewer 1 for the insightful comments. A revised version of the manuscript is uploaded. This addresses the major and minor points as follows:
Major comments
- The title of the manuscript is changed to "Areas of uncertainty in SARS-CoV-2 vaccination for cancer patients".
- The abstract is modified and now reads as "patients with cancer" rather than "patients with solid tumors and hematological malignancies". Data on patients with multiple sclerosis are briefly discussed only to examine the effect of B cell depleting therapies (anti-CD20) that are also used in hematological malignancies.
- This section is re-organized. In the revised version, 1) the uncertainties in humoral immunity, 2) cellular immunity and 3) cancer treatment modifications when there is a SARS CoV-2 infection in patients with cancer, are outlined for the discussion that follows.
- The discussion on a practical test that measures T cell immunity against SARS-CoV-2 for cancer patients is an important part of this review article. Similar tests have been proposed for CMV and VZV in vulnerable populations with defective humoral immunity. Serological tests are not always reliable to predict protection from vaccines in patients with cancer, e.g. those who receive treatment with B cell depleting drugs. Vaccine protection could inform modifications in cancer management, repeat vaccination and timing of vaccination in cancer patients. I agree that this is not unique to a particular infection, however the impact of SARS-CoV-2 on cancer patients compared to other infections makes the need for a test that measures T cell immunity to SARS-CoV-2 more pressing. In the revised manuscript, this section is incorporated in the previous section where T cell immunity is discussed. It is no longer outlined as a seperate point of discussion in the introduction. Also, the examples of CMV and VZV are mentioned with additional references.
- This section is dropped in the revised manuscript.
Minor comments
- A third person in past tense is used throughout in the revised manuscript to avoid confusion.
- These have been corrected. The term "index virus" has replaced the term "wild type virus" as per the WHO. Several other grammatical modifications are included in the revised manuscript.
- These have been corrected in the revised manuscript.
Reviewer 2 Report
This review deals with covid and the vaccination against it in the - heterologous - group of cancer patients, i.e. immunocompromised patients. As it sums up data on the most vulnerable group of people this is an important issue.
However, this is a broad overview on different topics concerning covid and cancer patients: besides vaccination also on diagnostic methods on determining vaccine efficacy in general, and also on morbidity and mortality of cancer patients under treatment and with treatment alterations due to covid. SARS-CoV2 vaccination is only one topic among others, so I doubt this article is placed well in "Vaccines".
Author Response
I would like to thank reviewer 2 for the critical review and feedback.
I believe that incorporation of a validated IGRA can inform future strategy to guide vaccination programs in cancer patients, where serological tests may not be accurate to predict protection. Discussion of topics about morbidity and mortality from Covid-19 in patients with cancer, is necessary to put vaccination in perspective. Also, it is currently unclear how, or whether to modify the treatment of a vaccinated cancer patient who is diagnosed with mild Covid-19 infection while undergoing radiation therapy or other cancer treatments. Raising this point as an area of uncertainty, where more research could be impactful is in my opinion within the scope of the journal and the special issue on Covid 19 and cancer.
Reviewer 3 Report
This manuscript is a review of the available data on the particular problems encountered in cancer patients for Covid-19. Where the cancer involves B cells, or in cases where the treatment in use affects B-cell function, the efficiency of vaccines is obviously reduced or absent, meaning that this group is at heightened risk of infection. Much the same is true of any other condition where immune function is moderated by treatment, lupus for example. But the review also describes the observations made over the pandemic period for other types of cancer including the general observations of collateral damage from lockdowns, which delayed diagnosis, treatment and surgical procedures, all of which contributed to higher mortality and morbidity than in the general population.
For the most part the review is well written but there are some aspects that need improvement.
-
The author should review the grammar usage throughout. The use of “we critically appraise” is generally when there are multiple authors although it can be interpreted as “we, the scientific body” but its use here is not clear as it is sole authorship. It would be much better to use the third person past tense throughout, to avoid any ambiguity.
-
The description of SARS-CoV-2 is limited. We have details of the spike interaction with ACE2 but little more except a later, short description of interferon antagonism. The VOCs are included but no biology is given. Omicron, for example, is attenuated and no longer uses the TMMPS entry mechanism described earlier. In consequence it is largely upper respiratory and does not cause the complication of lower RT infections. Because of this it is unclear to what extent the T-cell immunity suggested on lines 142-145 is relevant as the virus itself is not the same as the original emerging strain. That attenuation could be part of the lower M and M observed should be included.
-
The coverage of the vaccines is limited. S based vaccines are noted but nothing about the whole virus vaccines, which, in theory, should generate T-cell responses to N and other virus structural proteins as well as S. They were not popular early in the pandemic but more recent trials, VLA2001 for example, have been quite positive. There should be a nod toward these vaccines even if there is no direct data relating to cancer patient immunization.
-
An addition that should be made is that although the pandemic has given the opportunity to observe vaccine failure in cancer patients, it is not special for SARS-CoV-2. Any vaccine given to the same groups would also fail as the issue is their underlying immunity. I believe there is seasonal flu vaccine data that can be cited.
-
A brief explainer for the MAb and drugs mentioned should be included for readers not familiar with them. Currently, ocrelizumab is explained but alectinib and crizotinib are not.
Author Response
I would like to thank reviewer 3 for the constructive feedback. I have revised the manuscript to address the comments as follows:
- A third person in past tense is used throughout the revised manuscript to avoid confusion. Several other modifications in the revised manuscript address grammatical errors and improve clarity.
- The introduction is modified to clarify that there are 2 modes of viral entry, TMPRSS2 dependent and independent. Preference of the omicron for TMPRSS2 independent viral entry and how this affects the clinical syndrome is mentioned when the VOCs are described. The other structural and non-structural proteins beyond the spike are described.
- The inactivated whole virus vaccines are mentioned with references. A comment about their ability to raise immune responses against VOCs is added in lines 85-89 of the revised manuscript.
- This is an excellent comment. Data about the flu vaccination are referenced in the conclusion.
- Alectinib and crizotinib are described as "small molecule tyrosine kinase inhibitors".